# Impact of COVID-19 on patterns of drug utilization: A case study at national hospital

**Minh-Anh Le-Dang**[1], **Hai-Yen Nguyen-Thi** [1]*, **Luyen Pham Dinh**[1], **Danh Le Ngoc**[2], **Nguyen Dang Tu Le** [1,3], **Hien Pham Thu**[3], **Dinh Thanh Le**[3]*

**1** University of Medicine and Pharmacy at Ho Chi Minh City, Ho Chi Minh City, Viet Nam, **2** Ho Chi Minh City Department of Health, Ho Chi Minh city, Viet Nam, **3** Thong Nhat Hosital, Ho Chi Minh City, Viet Nam

\* haiyen@ump.edu.vn (H-YN-T); thanhld@bvtn.org.vn (DTL)

**Data Availability Statement:** The data utilized in this study were obtained from the electronic medical record database of Thong Nhat Hospital, located in Ho Chi Minh City, Vietnam. It is

## Abstract

### Background

The Coronavirus disease of 2019 (COVID-19) pandemic and the corresponding mitigation measures have had a discernible impact on drug utilization among outpatients. However, limited research exists on the prescription trends in the elderly population during the pandemic period in Viet Nam.

### Objectives

This study aims to analyze the effects of COVID-19 on outpatient drug utilization patterns at a national geriatric hospital in Ho Chi Minh City before and after the early onset of the pandemic.

### Methods

Data was collected from the prescriptions and administration claims, encompassing the period from January 2016 to December 2022. The dataset was divided into two periods: Period 1: January 2016 to December 2020 and Period 2: January 2021 to December 2022. The drug utilization was measured using DDD/1000P (defined daily doses–DDD per 1000 prescriptions) on a monthly basis. The analysis employed interrupted time series using Autoregressive Integrated Moving Average (ARIMA) to detect changes in drug use levels and rates.

### Results

A total of 1,060,507 and 644,944 outpatient prescriptions from Thong Nhat Hospital were included in Period 1 and Period 2, respectively. The median age of the patients were 58 in Period 1 and 67 years old in Period 2. The most common comorbidities were dyslipidemia, hypertension, and diabetes mellitus. In terms of medication utilization, cardiovascular drugs were the most frequently prescribed, followed by drugs active on the digestive and hormonal systems. The study observed significant surges in the number of prescriptions and the average number of drugs per prescription. However, there were no significant changes in the overall consumption of all drugs. Among the drug groups related to the cardiovascular

imperative to note that the data employed herein are of a strictly confidential nature and remain under the ownership of the hospital's governing authority. For inquiries regarding access to this data, kindly direct your correspondence to Ms. Tien Phung via email at khoaduoc@bvtn.org.vn.

**Funding:** The author(s) received no specific funding for this work.

**Competing interests:** The authors have declared that no competing interests exist.

system, three subgroups experienced a sudden and significant increase: cardiac therapy, beta-blocking agents, and antihypertensives, with increasing consumption levels of 1,177.73 [CI 95%: 79.29; 2,276.16], 73.32 [CI 95%: 28.18; 118.46], and 36.70 [CI 95%: 6.74; 66.66] DDD/1000P, respectively. On the other hand, there was a significant monthly decrease of -31.36 [CI 95%: -57.02; -5.70] DDD/1000P in the consumption of anti-inflammatory and antirheumatic products. Interestingly, there was a significant increase of 74.62 [CI 95%: -0.36; 149.60] DDD/1000P in the use of antigout preparations.

## Conclusion

COVID-19 resulted in a sudden, non-significant increase in overall drug consumption levels among outpatients. Notably, our findings highlight significant increases in the utilization of three drug groups related to the cardiovascular system, specifically cardiac therapy, beta-blocking agents, and antihypertensives. Intriguingly, there was a statistically significant increase in the consumption of antigout preparations, despite a decline in the monthly consumption rate of non-steroidal anti-flammatory drugs (NSAIDs). Further studies in the following years are necessary to provide a more comprehensive understanding of the impact of COVID-19 on outpatient drug utilization patterns.

## Introduction

In December 2019, a new strain of Coronavirus (SARS-CoV-2) causing acute respiratory infections was identified in Wuhan City, China [1]. This disease, later named COVID-19, rapidly spread worldwide, resulting in severe consequences for healthcare systems in various countries. To address the complex epidemic situation, many countries implemented control measures aimed at mitigating the spread of COVID-19 within communities [2,3]. The pandemic had a significant impact on several hospital services, particularly outpatient care. During the initial stages of the outbreak, there was a notable decline in outpatient visits due to curfews and concerns about infection in public spaces [4–6]. Additionally, the global disruption of supply chains caused by COVID-19 has posed a serious challenge for patients who are relying on long-term medication use, impeding their access to prescription drugs [4,7].

Viet Nam swiftly implemented social distancing and other control measures to minimize virus transmission as soon as community transmission cases emerged [8]. After three phases of the epidemic, Viet Nam only recorded 35 deaths and 1,564 community cases, as deaths were elderly and/or comorbid patients [9]. However, the highly contagious Delta variant led to a surge in positive COVID-19 cases and deaths during the fourth wave of the pandemic. Especially Ho Chi Minh City, the largest city in Viet Nam, became an epicenter with a daily influx of new cases and deaths, particularly thousands of cases and hundreds of deaths per day, putting a strain on the healthcare system [9,10]. Patients who had positive COVID-19 test were forced to stay in field hospitals, which led to an overload of facilities and work for healthcare workers. In response to this crisis, the People's Committee of Ho Chi Minh City enforced a strict social distancing order. People were asked to stay at home and only go outside for essential purposes (buying food, medicines, or emergency care). They were not allowed to gather more than two people in public. This order's duration was extended; hence, Ho Chi Minh City went through a near 4-month lockdown with several rigorous measures [11].

COVID-19 has disrupted both individual visits and access to medical prescriptions from outpatient services at hospitals, which caused many disadvantages in regards to worsen

symptoms of chronic diseases [4,12–14]. Several studies have examined the impact of COVID-19-related restrictions on drug utilization among outpatients, revealing fluctuations in the prescription patterns for chronic conditions [15–17]. However, there is a limited body of research focusing on the elderly population, who require special attention due to their high prevalence of chronic diseases and comorbidities, making them particularly vulnerable to the effects of the pandemic [18–20]. Furthermore, as COVID-19 continues to evolve and new cases continue to rise [21], there is a critical need for studies investigating the impact of the pandemic on drug utilization trends among elderly outpatients. This study aims to analyze the effects of COVID-19 on outpatient drug utilization patterns in a geriatric hospital in Ho Chi Minh City before and after the early onset of the pandemic.

## Methods

### Study setting and design

A longitudinal single-center study was used to evaluate the impact of COVID-19 on the utilization trends of selected drugs among outpatients. The study conducted in Thong Nhat hospital located in Ho Chi Minh City, which is one of the largest national geriatric hospitals and the largest geriatric hospital in Viet Nam and the Southern Viet Nam, respectively. Data were derived from electronic medical database owned by Thong Nhat hospital. During the fourth wave of COVID-19, Thong Nhat hospital was in charge of a 1000 bed multi-class COVID-19 field hospital named Tan Binh district field hospital. Although there was a surge in new cases of COVID-19 during our study period, outpatient data in Thong Nhat hospital would not recorded a great number of COVID-19 confirmed patients since these patients were gathered and treated in the field hospital.

We evaluated the early impact of COVID-19 on the drug utilization. Ho Chi Minh city was effectively controlled the pandemic until December 2020 before positive COVID-19 cases rose again at the beginning of 2021. The city was catastrophically impacted by the fourth wave (from April 27th 2021) pandemic outbreak compared to previous waves; therefore, we assessed the impact of COVID-19 focusing on 2021. The first community case of 2021 in Ho Chi Minh City was reported at the end of January 2021, and a community case with unknown source of infection was reported on February 2021, subsequently [22,23]. Due to the risk of COVID-19 outbreak when people from other provinces and cities returning after holiday (Reunification Day on April 30th and Labour day on May 1st), Ho Chi Minh City issued retrictions [24]. However, the measures did not effectively combat the community spread; therefore, new COVID-19 cases rapidly increased at the initial stage of the fourth wave. The People's Commitee of Ho Chi Minh City raised several levels of social distancing order and extended the duration of these measures from June 1st 2021 to September 30th 2021 in accordance with a high peak on this pandemic wave. On October 1st 2021, Ho Chi Minh City was implemented a loose strategy to gradually lift the strict social distancing order (Fig 1) [25]. In regard to ensure healthcare for citizens during the social distancing period, the Viet Nam Minister of Health has imposed a temporary policy for drug use in outpatient with chronic disease in February 2021 allowing patients who are on long-term drugs to claim a three-month supply to limit the frequency in visiting hospitals [26]. Additionally, there have not been telemedicine deployed in Viet Nam, and patients are unable to claim drug refills in other nearer healthcare departments [27]. Therefore, to refill their prescription drugs, patients have to visit their hospitals in-person. Consequently, we decided to select the intervention time point was January 2021 to capture the impact of COVID-19 broadly and early on drug utilization. The study was divided into two periods: (i) Period 1: from January 2016 to December 2020; (ii) Period 2: the early onset of COVID-19: January 2021 to December 2022).

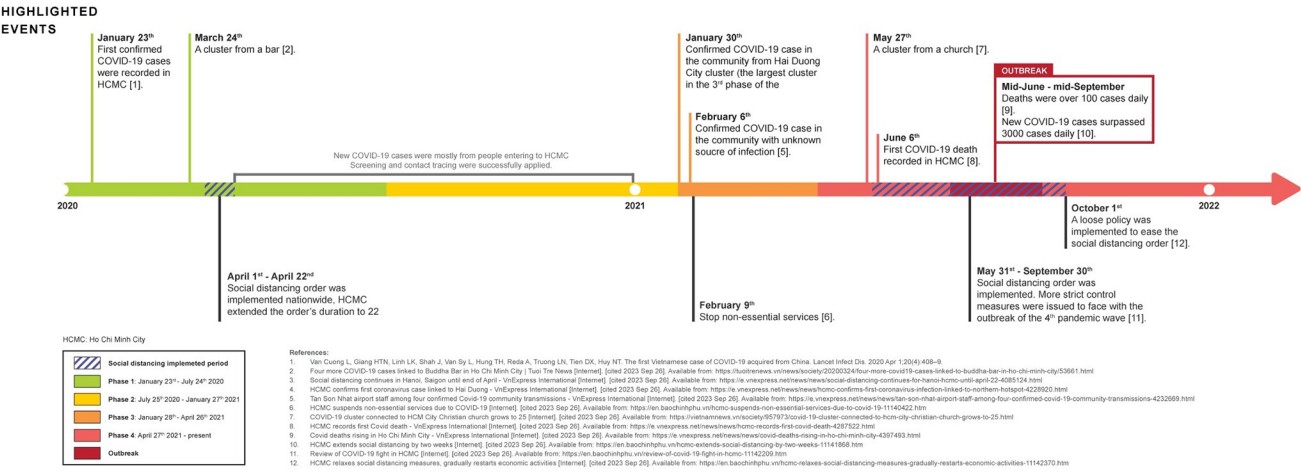

**Fig 1. The timeline of COVID-19 in Ho Chi Minh City.**

We extracted the patient-level data from the prescriptions and administration claims of the hospital in the periods of January 2016 and December 2022. We excluded patients under 18 years old, prescriptions contained only herbal medicines (Fig 2).

## Study outcomes

We estimated drug consumptions by measuring DDD/1000P calculated by dividing the total DDD per month by the number of prescriptions per month then multiplying by 1000. We examined the monthly DDD/1000P of the following medication classes: (i) Drugs active on cardiovascular system: cardiac therapy, antihypertensives, diuretics, beta blocking agents, calcium channel blockers (CCB), agents acting on the Renin-Angiotensin system (RAASi), lipid modifying agents; (ii) Drugs active on digestive system: acid related disorders, functional gastrointestinal disorders; (iii) Drugs active on hormonal system: corticosteroids for systemic use; thyroid therapy, drugs used in diabetes; (iv) Analgesics, antipyretics, NSAIDs and other drugs related to arthritis: immunosuppressants; anti-inflammatory and antirheumatic products; antigout preparations; drugs for treatment of bone diseases; analgesics; (v) Antimicrobials: antibacterials for systemic use (Antibiotics), antivirals for systemic use (Antivirals). Additionally, we also examined all drugs which have available ATC codes and DDD to assess the consumption of all drugs.

Furthermore, we examined the monthly number of outpatients that statisfied the inclusion criteria and the monthly average number of drugs per prescription. In addition, to generally describe the population in two periods, we provide patient charateristics encompassing age, cormobidities, Charlson Comorbidity index. In terms of drug consumption, we provide average number of drugs per precriptions, the most use drug groups based on its pharmacology, and the consumption of drugs classified by ATC code level 01 measured by DDD/1000P.

## Statistical analysis

Interrupted time series design has been the most robust method in quasi-experiment designs, which is commonly used to assess the impact of a specific intervention or a program in healthcare research [28]. There are three models that commonly use in interrupted time series method, which are linear regression, segmented linear regression and ARIMA. In this study,

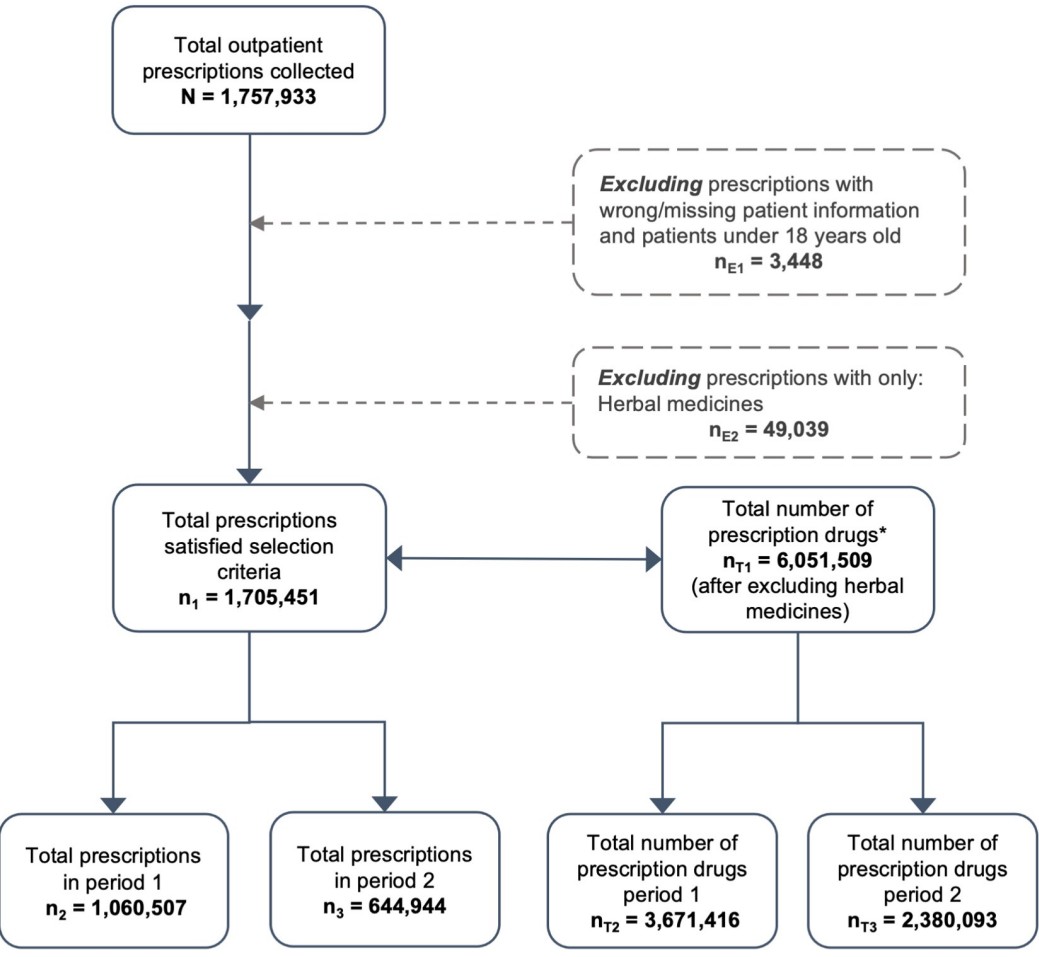

**Fig 2. Data collection flow.** *The total number of drugs calculating from the total prescriptions satisfying selection criteria (e. g. 2 prescriptions were collected: prescription 1 has 3 drugs prescribed, prescription 2 has 5 drugs prescribed; therefore, 8 is the total number of prescription drugs).*

we apply ARIMA to evaluate the impact of COVID-19 on the drug utilization with the intervention point in January 2021. ARIMA can use for dependent and correlative residuals, which does not meet the assumptions of linear regression model. Especially, it is more feasible than segmented linear regression model to form algorithms for the self-correlation, non-stationary, and seasonality factors, which usually occurs in time series data (mostly affected by time) [29,30].

In order to apply the most suitable ARIMA model, we used the auto arima function and revised by Ljung-Box test. If the p-value of Ljung-Box test above 0.05, the residuals are the white noise; thereby, the chosen model would be appropriate. ManKendall test was used to examine the trend in time series (p-value < 0.05), and Augmented Dickey-Fuller test was used to examine the stationary of time series (p-value < 0.05). The effect of COVID-19 will be evaluated by two variables: change in level (step change) and change in slope (ramp). "Step change" is an immediate, sustained change (up or down) which we can observe starting from the time where the intervention applied. "Ramp" is a change in slope which immediately starts from the intervention [29].

We used the model to detect the step change and ramp monthly in the number of prescriptions, the average number of drugs per prescription and the drug consumptions, compared to their counterfactuals [29,31,32]. Analysis was conducted using R language version 4.3.1 integrated with Rstudio ver 2022.12.0+353.

## Ethical consideration

This study and data access has been approved by the Thong Nhat hospital Research Ethics Committee. Consent was not required as data are de-identified. The data was accessed and collected from January to March 2023.

## Results

### Characteristics of the data population

A total of 1,060,507 and 644,944 outpatients' prescriptions at Thong Nhat Hospital were included in this study during Period 1 (January 2016 to December 2020) and Period 2 (January 2021 to December 2022), respectively. In the first period, the percentage of patients aged from 30 to below 60 was the highest prevalence. However, the age group accounted for the highest proportion in period 2 was the elderly group (from 60 years old and older). The median age in Period 1 and Period 2 were 58 (46–68) and 67 (58–76) years old, respectively. The analysis of diagnosed diseases revealed noteworthy changes. The proportion of individuals with a single diagnosed disease decreased from 23.74% (Period 1) to 10.85% (Period 2). While the prevalence of individuals with multiple diagnosed diseases, notably six or more, increased substantially from 1.48% to 15.05%. The CCI was distributed similarly between the two periods. However, the prevalence of individuals with no comorbidities decreased slightly from 85.38% (Period 1) to 82.18% (Period 2). The most common comorbidities were dyslipidemia, hypertension, and diabetes mellitus. There were notable increases in the prevalence of GERD (from 3.29% to 5.10%) and Angina pectoris (from 1.04% to 5.32%) Table 1.

The average number of drugs per prescriptions in Period 1 and 2 were 3.46 (1.73) and 3.69 (1.97), respectively. The proportion of prescriptions under 05 drugs dominated in both periods, accounting for 75.04% and 68.29%, respectively. However, there was a slightly increased noticed in the percentage of prescriptions with 5 or more drugs, from 24.96% (Period 1) to 31.71% (Period 2). Regarding medication utilization, cardiovascular drugs were the most used drug group, followed by drugs active on the digestive and hormonal systems. Notable changes were observed in drug consumption rates (DDD/1000P), with varying trends across different drug categories Table 1.

### Interrupted time series analysis

During the initial stage of Period 2, we observed a sudden and significant increase in the number of outpatients, as indicated by the number of prescriptions. The number of patients surged to 19,425 (p < 0.001), with a non-significant decrease of 10 prescriptions (p = 0.987) monthly. Additionally, we found a significant increase of 0.40 drugs per prescription (p = 0.007), with a non-significant monthly decrease of 0.001 drugs per prescription (p = 0.888) S3 Table.

Overall, there was no significant change in the overall level of drug consumption during the early stage of Period 2. Specifically, the change in consumption level (ramp) was +153,201.70 [CI 95%: -13,032.08; 319,435.5] DDD/1000P, and the change in consumption rate (step change) was -1,911.52 [CI 95%: -38,728.51; 34,905.47] DDD/1000P per month Table 2 and S4 Table.

**Table 1. Characteristics of the data population at Thong Nhat hospital from January 2016 to December 2022 by study defined period.**

| | Period 1 | Period 2 |
|---|---|---|
| **Age** (years), median (Q1-Q3) | 58 (46–68) | 67 (58–76) |
| <30, % | 6.72 | 3.17 |
| 30–60, % | 48.16 | 25.42 |
| >60, % | 45.12 | 71.42 |
| **Charlson Comorbidity Index (CCI)**, % | | |
| 0 | 85.38 | 82.18 |
| 1 | 13.10 | 15.51 |
| 2 | 1.42 | 1.99 |
| ≥ 3 | 0.10 | 0.32 |
| **Number of diagnosis diseases**, % | | |
| 1 | 23.74 | 10.85 |
| 2 | 21.03 | 14.13 |
| 3 | 17.81 | 15.48 |
| 4 | 18.12 | 21.52 |
| 5 | 17.82 | 22.96 |
| ≥6 | 1.48 | 15.07 |
| **Comorbidities**, % | | |
| Dyslipidemia | 14.47 | 15.05 |
| Hypertension | 14.33 | 13.91 |
| Diabetes mellitus | 7.90 | 7.92 |
| Ischemic heart disease | 7.39 | 5.77 |
| GERD | 3.29 | 5.10 |
| Angina pectoris | 1.04 | 5.32 |
| **Drug groups**[a], % | | |
| Drugs active on cardiovascular system | 41.16 | 50.96 |
| Drugs active on digestive system | 29.45 | 25.64 |
| Drugs active on hormonal system | 27.46 | 25.54 |
| Analgesics, antipyretics, NSAIDs and other drugs related to arthritis | 21.71 | 19.48 |
| Antimicrobials | 18.22 | 13.60 |
| **Number of drugs per prescription**, mean (SD) | 3.46 (1.73) | 3.69 (1.97) |
| <5, % | 75.04 | 68.29 |
| ≥5, % | 24.96 | 31.71 |
| **Drug consumptions**[b], DDD/1000P | | |
| A | 55125.58 | 58078.64 |
| B | 5319.60 | 8895.53 |
| C | 25491.52 | 44332.56 |
| D | 15.13 | 37.83 |
| G | 1019.44 | 3124.80 |
| H | 854.12 | 753.37 |
| J | 3213.52 | 1920.24 |
| L | 26.45 | 16.07 |
| M | 2797.27 | 2466.18 |
| N | 6327.48 | 6094.34 |
| P | 25.18 | 9.12 |
| R | 5847.65 | 2081.24 |
| S | 3.20 | 3.07 |

*(Continued)*

**Table 1.** (Continued)

| | Period 1 | Period 2 |
|---|---|---|
| V | 0.09 | 0.04 |

[a]We present the common variables (top 5) with the highest number of prescriptions, please refer to S1 Table for full results of the number of prescriptions by pharmacological group.

[b]Please refer to S2 Table for drug consumptions (DDD/1000P) corresponding with ATC code level 3 in 2 periods.

Furthermore, we observed varying changes in drug utilization across different subgroups during Period 2, Table 2. The consumption level of drug subgroups related to the cardiovascular system showed increases in level, with most of them (6/8 subgroups) increasing monthly on average. However, the changes in monthly consumption rate were not significant S1 Fig.

**Table 2. Interrupted time series using autoregressive models.**

| Classification | ATC code level 03 | Drug group | Change in level | 95% CI | p | Change in slope | 95% CI | p |
|---|---|---|---|---|---|---|---|---|
| All drugs | | | 153200.0 | [-13032.08;319435.5] | 0.071 | -1911.50 | [-38728.51; 34905.47] | 0.919 |
| Drugs active on cardiovascular system | C01 | **Cardiac therapy** | **1177.73** | **[79.29; 2276.16]** | **0.036** | -45.57 | [-269.79; 178.64] | 0.690 |
| | C02 | **Antihypertensives** | **36.70** | **[6.74; 66.66]** | **0.016** | 0.35 | [-1.71; 2.42] | 0.737 |
| | C03 | Diuretics | 48.59 | [-112.70; 209.88] | 0.555 | 2.18 | [-13.60; 17.96] | 0.786 |
| | C07 | **Beta-blocking agents** | **73.32** | **[28.18; 118.46]** | **0.001** | -0.11 | [-4.25; 4.04] | 0.960 |
| | C08 | CCB | 685.05 | [-852.23; 2222.34] | 0.382 | 61.59 | [-252.21; 375.39] | 0.701 |
| | C09 | RAASi | 2054.46 | [-1702.50; 5811.43] | 0.284 | 127.78 | [-639.11; 894.66] | 0.744 |
| | C10 | Lipid modifying agents | 2380.38 | [-2336.90; 7097.65] | 0.323 | 77.00 | [-697.33; 851.33] | 0.846 |
| | B01 | Antiplateles[a] | 1931.03 | [-336.87; 4198.93] | 0.095 | 47.62 | [-415.32; 510.55] | 0.840 |
| Drugs active on digestive system | A02 | Drugs for acid related disorders | -1998.23 | [-6807.26; 2810.81] | 0.415 | 395.10 | [-257.22; 1047.43] | 0.235 |
| | A03 | Drugs for functional gastrointestinal disorders | -182.41 | [-656.19; 291.36] | 0.450 | -2.82 | [-59.86; 54.22] | 0.923 |
| Analgesics, antipyretics, NSAIDs and other drugs related to arthritis | L04 | Immunosupressants | -12.00 | [-34.48; 10.55] | 0.298 | -1.40 | [-3.12; 0.32] | 0.110 |
| | M01 | **Antiinflammatory and antirheumatic products** | 213.68 | [-203.08; 630.45] | 0.315 | **-31.36** | **[-57.02; -5.70]** | **0.017** |
| | M04 | **Antigout preparations** | **74.62** | **[-0.36; 149.60]** | **0.051** | 8.51 | [-6.79; 23.82] | 0.276 |
| | M05 | Drugs for treatment of bone diseases | 53.15 | [-142.28; 248.58] | 0.594 | -2.44 | [-26.26; 21.39] | 0.841 |
| | N02 | Analgesics | 359.07 | [-1506.15; 2224.30] | 0.706 | -11.01 | [-269.58; 247.55] | 0.933 |
| Hormone and drugs active on hormone system | A10 | Drugs used in diabetics | 2382.22 | [-17319.35; 22083.80] | 0.813 | 1369.34 | [-2652.26; 5390.93] | 0.505 |
| | H02 | Corticosteroids for systemic use | -115.18 | [-297.51; 67.15] | 0.216 | 0.70 | [-12.24; 13.63] | 0.916 |
| | H03 | Thyroid therapy | -34.44 | [-106.46; 37.58] | 0.349 | -0.11 | [-5.55; 5.33] | 0.969 |
| Antimicrobials | J01 | Antibacterials for systemic use | -46.94 | [-868.71; 774.82] | 0.911 | -26.09 | [-113.44; 61.26] | 0.558 |
| | J05 | Antivirals for systemic use | -132.90 | [-336.82; 71.01] | 0.201 | 3.10 | [-38.52; 44.72] | 0.884 |

[a]Drug group with ATC code level 3: B01 in drugs active on cardiovascular system is antiplatelets.

Three subgroups that experienced a sudden and significant increase were cardiac therapy, beta-blocking agents, and antihypertensives, (Fig 3) with increasing consumption levels of 1,177.73 [CI 95%: 79.29; 2,276.16], 73.32 [CI 95%: 28.18; 118.46], and 36.70 [CI 95%: 6.74; 66.66] DDD/1000P, respectively.

Regarding drug groups related to the digestive system, there were non-significant declines in consumption level during Period 2, S2 Fig. The use of drugs for acid-related disorders increased 395.10 [CI 95%: [-257.22; 1047.43] DDD/1000P monthly, while drugs for functional gastrointestinal disorders decreased -2.82 [CI 95%: -59.86; 54.22]. Drugs affecting the hormonal system showed an immediate but non-significant decrease on average in usage, including systemic corticosteroids and thyroid-related drugs, except for antidiabetic drugs, which had a sudden increase at the beginning of Period 2 and continued to increase monthly, although non-significantly S3 Fig.

In terms of two commonly prescribed drug groups for acute conditions, we observed a significant monthly decrease of -31.36 [CI 95%: -57.02; -5.70] DDD/1000P in the use of anti-inflammatory and antirheumatic products, alongside a non-significant increase in consumption level (Fig 3). Conversely, there was a significant increase of 74.62 [CI 95%: -0.36; 149.60] DDD/1000P in the use of antigout preparations, with a further non-significant increase in consumption rate (Fig 3). Other groups such as analgesics and drugs for treatment of bone diseases showed non-significant positive changes in consumption level, except for immunosuppressants, which experienced a sudden decrease. And those subgroups non-significantly decreased monthly in consumption rate during the observation period S4 Fig.

In regard to antimicrobials, there was a non-significant negative change in level witnessed in both subgroups, which are antibiotics and antivirals. The use of antibiotics further non-significantly reduced monthly, in contrast, antivirals experienced a non-significant increase in slope S5 Fig.

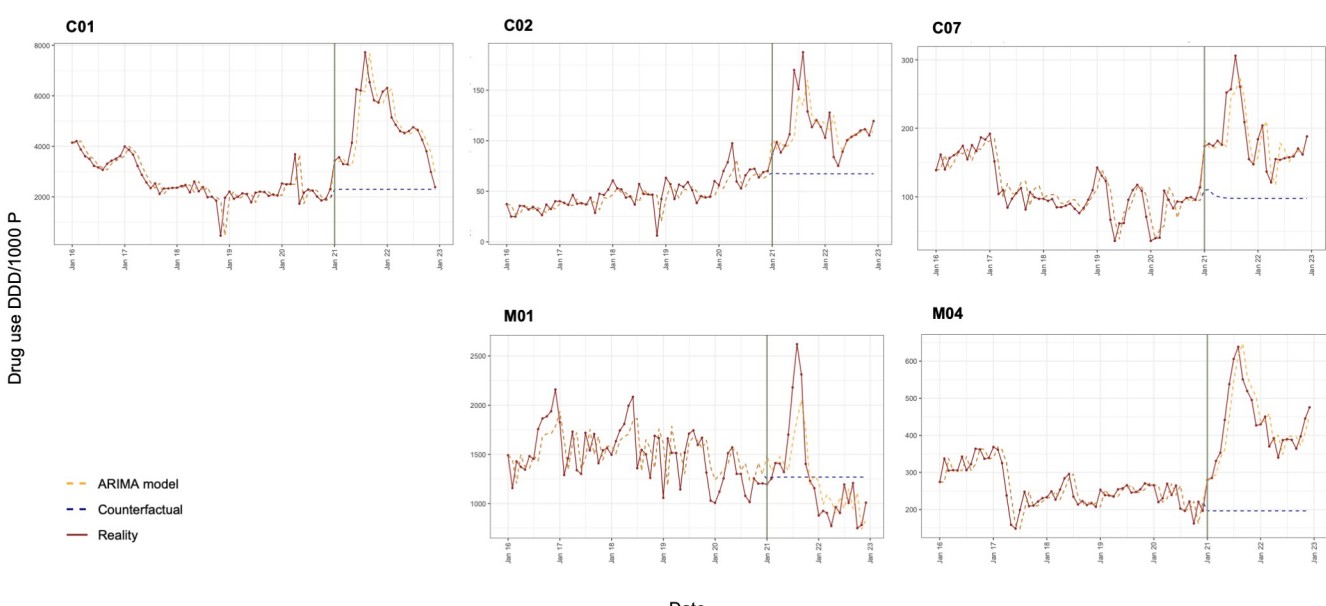

**Fig 3. Drug use monthly measured by DDD/1000P in (C01) Cardiac therapy, (C02) Antihypertensives, (C07) Beta-blocking agents, (M01) Antiinflammatory and antirheumatic products, and (M04) Antigout preparations from January 2016 to December 2022.**

## Discussion

COVID-19 has significantly disrupted the treatment of patients with multiple chronic conditions [4,12–14]. In many countries, telemedicine services have been widely adopted to ensure a continuous drug supply while minimizing the risk of COVID-19 transmission [33]. However, Viet Nam has not implemented telemedicine services as extensively [27]. In our observations, we found that patients, especially those with chronic diseases, resorted to psychological stockpiling of drugs as a coping mechanism during the prolonged period of social distancing and restrictions imposed by the pandemic. In order to prepare before the stay-at-home orders, patients in some US states having US insurance were accepted to claim extra drugs early, in turn leading to a surge in refilling prescriptions at the onset of COVID-19 [17]. Unfortunately, due to confidentiality issues in accessing databases, we were unable to quantitatively interpret the extent of psychological stockpiling of drugs among hospital patients. This situation underscores the need for further investigation and the prompt development of a comprehensive strategy for managing chronic conditions in the event of similar situations, such as future outbreaks like COVID-19.

Regarding the consumption of all drugs, there was a non-significant increase during the onset of COVD-19. Among drug groups used for the treatment of chronic diseases, cardiovascular drugs group was the only group noticed an immediately significant change in level, particularly cardiac therapy, beta-blocking agents and antihypertensives. Note that outpatients in Viet Nam typically obtain their refilled prescriptions from the hospital where they have already registered in the social health insurance. Besides, the results showed a high proportion of chronic diseases, which requires periodical examinations, and increases in the consumption of cardiovascular drugs from Period 1 to Period 2 were driven by the surge of it in the initial stage of the COVID-19 outbreak, which has been explained by the abovementioned stockpiling behavior. Therefore, the study population, especially during the transition from Period 1 to Period 2, mainly comprised outpatients who have already had long-term follow-up at the hospital. Consequently, based on outpatients followed up for long periods, it is anticipated that there might be deterioration in the cardiovascular conditions or the appearance of additional cardiovascular diseases, taking the increases in the consumption of cardiovascular drugs into consideration. A study on older Italian population by Marengoni et al showed that the percentage of drugs active on the cardiovascular system use fluctuated in all elderly population aged sub-groups ranging from -21.2 to 3.6% in COVID-19 period compared to pre-pandemic period with an increase observed in 65–69 aged, while there were positive changes in the consumption of antihypertensives during COVID-19 pandemic in 65–69 aged group and 75–79 aged group, with an increase by 2.4% and 0.7%, respectively [15]. In contrast, the drug use of outpatients with Medicaid by Elizabeth Williams experienced a decrease of 3% in the drug utilization of cardiac drugs in COVID-19 period (2020) compared to the pre-COVID-19 period (2019) [34]. However, with the influence of COVID-19 restrictions and curfew, a study by Aboulatta et al in Canadian citizens noted a sharply decline in the consumption of cardiovascular drugs in incident users of -23.05% immediately after enacting COVID-19 restriction, followed by a rebound increase monthly, while in prevalent users only observed a gradual drop monthly in the use of cardiovascular drugs [16].

With the drug utilization in subgroups commonly used for acute treatment (analgesic, antipyretic, NSAIDs, other drugs related to arthritis and antimicrobials) regarding the early stage of COVID-19, interestingly, there was a sudden significant in the use of antigout preparations. On the contrary, we noticed a significantly drop in the anti-inflammatory and antirheumatic products consumption monthly. During COVID-19 period, even without restrictions or curfews, many people appeared not to visit hospitals in-person if they had common mild

symptoms and/or did not require urgent procedures due to the encouragement from the healthcare authorities as well as the fear of infection. Therefore, it might be one of the reasons explained for a decline in anti-inflammatory and antirheumatic products consumption monthly. Marengoni et al noted an increase in drugs for gout treatment in patients from 65–79 years old, with a range from 0.3–7.3% compared to pre-COVID-19. A similar pattern with our result in NSAIDs consumption was also found in Marengoni et al's study, which noted decreases in the proportions of NSAIDs consumption in all elderly aged sub-groups ranging from -5.8 to -25.3% compared to pre-COVID-19 [15]. Medicaid users also noted a negative change in the percentage of analgesics/antipyretics use, with -6% compared pre-pandemic period with COVID-19 period [34]. Being impacted by COVID-19 mitigation measures, study in Manitoba, Canada by Aboulatta et al noted a sudden significantly drop by 0.4289% and 0.5175% in the NSAIDs use of incidence users and prevalent users, respectively; however, there was no monthly further reduction observed in this drug [16]. Other acute drug subgroup, which is antimicrobials, observed a downward trend throughout the study period, which appeared to be effectively impacted by antimicrobial stewardship program applied in the hospital, and the disease patterns are mainly chronic diseases. Even though there were no statistical changes in the use of antimicrobials, we still noted that the use of this group non-significantly decreased, while the antivirals use experienced a gradual increase. The percentage of antibiotics use in older population also experienced a drop from 19.7–27.7% in Marengoni et al study [15]. The use of antibiotics in Medicaid population drastically decreased by 23% in compared to pre-COVID-19 year [34]. In addition, the impact of COVID-19 curfews and hygiene measures such as wearing face masks, washing hands frequently, sanitizing have showed an effectiveness on reducing patients with other communicable diseases as well [35–37]. Consequently, these control measures have contributed to sudden declines in the consumption of antibiotics in outpatients during the COVID-19 period, which were witnessed in several studies [38–40].

COVID-19 has significantly changed in drug utilization trend, especially in outpatient prescriptions. With the circumstance of interrupted drug supply chains, it is crucial to analyze future drug use trends in the following years with the impact of COVID-19 in order to support healthcare policymakers implement appropriate measures. Therefore, drug products could be ensured in terms of volume to supply for patients with a careful attention on elderly population, who are commonly diagnosed with chronic conditions and have the high proportions of comorbidities in associated with a long-term medication use.

This study has several limitations. Firstly, due to confidentiality concerns, we were unable to access additional personal information of the outpatients. As a result, the detailed characteristics of the patients could not be fully captured for further analysis. Secondly, each prescription in the study had its own unique ID number, making it impossible to differentiate between prevalent and incident users of the medications. Thirdly, a proportion (30.59%) of the active ingredients commonly used in the hospital lacked corresponding ATC codes and DDD values. This limitation restricted our ability to accurately analyze drug consumption trends during the study period. Therefore, it is important to consider those limitations when interpreting the findings of this study.

## Conclusion

COVID-19 led to a sudden non-significantly increase in overall drugs consumptions level. Additionally, our findings highlight the significant increases in the use of three groups in the drugs active on cardiovascular system, which are cardiac therapy, beta-blocking agents and antihypertensives. Interestingly, we found a statistically significant increase in consumption

level of antigout preparations, contrary to the decline on monthly consumptions rate of NSAIDs. Further studies in the following years are needed to provide a better scenario of COVID-19 impact on outpatient's drug utilization patterns.

## Supporting information

**S1 Fig. Interrupted time series analysis of monthly drug consumption measured by DDD/ 1000P in: (A) Drugs active on cardiovascular system.**
(TIF)

**S2 Fig. Interrupted time series analysis of monthly drug consumption measured by DDD/ 1000P in: (B) Drugs active on digestive system.**
(TIF)

**S3 Fig. Interrupted time series analysis of monthly drug consumption measured by DDD/ 1000P in: (D) Hormon and drugs active on endocrine system.**
(TIF)

**S4 Fig. Interrupted time series analysis of monthly drug consumption measured by DDD/ 1000P in: (C) Analgesics, antipyretics; non-steroidal anti-inflammatory drugs and other drugs related to arthritis.**
(TIF)

**S5 Fig. Interrupted time series analysis of monthly drug consumption measured by DDD/ 1000P in: (E) Antimicrobials.**
(TIF)

**S1 Table. Number of prescriptions by pharmacological group.**
(DOCX)

**S2 Table. Drug consumptions categorize by ATC code level 03 (DDD/1000P).**
(DOCX)

**S3 Table. The number of prescriptions' data.**
(DOCX)

**S4 Table. Final fitted ARIMA Models provides the model specifications for the main ARIMA analyses presented in Table 2.**
(DOCX)

## Author Contributions

**Conceptualization:** Minh-Anh Le-Dang, Hai-Yen Nguyen-Thi, Luyen Pham Dinh, Danh Le Ngoc, Nguyen Dang Tu Le, Hien Pham Thu, Dinh Thanh Le.

**Data curation:** Minh-Anh Le-Dang, Hai-Yen Nguyen-Thi, Nguyen Dang Tu Le, Dinh Thanh Le.

**Formal analysis:** Minh-Anh Le-Dang, Hai-Yen Nguyen-Thi, Danh Le Ngoc, Nguyen Dang Tu Le, Hien Pham Thu, Dinh Thanh Le.

**Investigation:** Minh-Anh Le-Dang, Hai-Yen Nguyen-Thi, Luyen Pham Dinh, Nguyen Dang Tu Le, Hien Pham Thu.

**Methodology:** Minh-Anh Le-Dang, Hai-Yen Nguyen-Thi, Luyen Pham Dinh, Nguyen Dang Tu Le, Hien Pham Thu, Dinh Thanh Le.

**Project administration:** Hai-Yen Nguyen-Thi, Luyen Pham Dinh, Nguyen Dang Tu Le.

**Resources:** Hai-Yen Nguyen-Thi, Nguyen Dang Tu Le, Hien Pham Thu, Dinh Thanh Le.

**Software:** Minh-Anh Le-Dang, Nguyen Dang Tu Le.

**Supervision:** Minh-Anh Le-Dang, Hai-Yen Nguyen-Thi, Luyen Pham Dinh, Nguyen Dang Tu Le, Dinh Thanh Le.

**Validation:** Minh-Anh Le-Dang, Hai-Yen Nguyen-Thi, Luyen Pham Dinh, Danh Le Ngoc, Nguyen Dang Tu Le, Hien Pham Thu, Dinh Thanh Le.

**Visualization:** Minh-Anh Le-Dang, Nguyen Dang Tu Le.

**Writing – original draft:** Minh-Anh Le-Dang, Hai-Yen Nguyen-Thi, Luyen Pham Dinh, Danh Le Ngoc, Nguyen Dang Tu Le, Hien Pham Thu, Dinh Thanh Le.

**Writing – review & editing:** Minh-Anh Le-Dang, Hai-Yen Nguyen-Thi, Luyen Pham Dinh, Danh Le Ngoc, Nguyen Dang Tu Le, Hien Pham Thu, Dinh Thanh Le.

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
