## [Decision Letter · Decision Letter 0]

11 Sep 2023

PONE-D-23-22693Impact of COVID-19 on patterns of drug utilization: a case study at national hospitalPLOS ONE

Dear Dr. Nguyen,

Thank you for submitting your manuscript to PLOS ONE. After careful consideration, we feel that it has merit but does not fully meet PLOS ONE’s publication criteria as it currently stands. Therefore, we invite you to submit a revised version of the manuscript that addresses the points raised during the review process.

We look forward to receiving your revised manuscript.

Kind regards,

Chinh Quoc Luong, MD., PhD.

Academic Editor

PLOS ONE

Journal Requirements:

2. Please ensure that you refer to Figure 2 in your text as, if accepted, production will need this reference to link the reader to the figure.

Additional Editor Comments:

The manuscript submitted by the authors has undergone peer review by two experts in the field. I have received two completed reviews, and the reviewers’ comments are available below. The manuscript presents a longitudinal single-center study that analyzes the effects of COVID-19 on outpatient drug utilization patterns at a national geriatric hospital in Ho Chi Minh City before and after the early onset of the pandemic in Vietnam. The topic studied by the authors is of interest to readers of PLoS ONE. However, the reviewers have raised several concerns that should be addressed. Based on their comments, I invite the authors to submit a revised version of their paper that addresses the points raised during the review process.

Reviewers' comments:

Reviewer's Responses to Questions

**Comments to the Author**

1. Is the manuscript technically sound, and do the data support the conclusions?

Reviewer #1: Partly

Reviewer #2: Yes

2. Has the statistical analysis been performed appropriately and rigorously? 

Reviewer #1: No

Reviewer #2: Yes

3. Have the authors made all data underlying the findings in their manuscript fully available?

Reviewer #1: Yes

Reviewer #2: No

4. Is the manuscript presented in an intelligible fashion and written in standard English?

Reviewer #1: Yes

Reviewer #2: Yes

5. Review Comments to the Author

Reviewer #1: Please see attachment with my complete review. Regarding my comment that the statistical analysis is not as rigorous as it could be, I think the issues are relatively feasible to resolve.

Best of luck with the project!

Reviewer #2: This is a great paper assessing an important area on the impact of COVID-19 on drug utilization in a national hospital in Viet Nam. The manuscript is overall well written and easy to follow. Overall, well done! I have some comments for the authors below;

•In the results section of the abstract, authors state “The median age of the patients ranged from 58 to 67 years old” . Do they mean the median in period 1 was 58 and and 67 in period 2? Authors could clarify this as median is one value and not a range. Authors could provide the interquartile range for each median value.

•Authors could use the full name of the month e.g., December instead of Dec for consistency throughout the document.

•Could the authors provide a table or figure of the timeline of the interventions/strategies implemented by the government/ministry of health to reduce the spread of COVID-19?

•Is there a justification or reason for excluding those aged below 18?

•Could the authors add on Table 1 the missing data for each characteristic to help the reader know what proportion of data was missing for each characteristic as I understand the analysis was done for complete data a.k.a Complete Case Analysis (CCA).

•Were any of the patient characteristics such as age, comorbidities etc accounted for in the interrupted time series model?

•Authors used ARIMA model to fit the ITS models, could Autocorrelation and partial autocorrelation function (ACF and PACF) plots be provided in the supplementary materials.

•Authors could reference the R packages and also the R software used for the ARIMA model if possible as it’s good practice to recognize these softwares

•In the sentence “the average number of drugs per prescription and the drug consumptions, compared to their counterfactuals in order to assess.[22–24]”. Seems something is missing at the end of the sentence? In order to assess what?

•In the sentence . In the first period, the 30- to below 60-year-old was the highest prevalence. Do authors mean the majority of the patients were aged 30-60 in your study dataset? Maybe rewording of the sentence could make it clearer

•Authors could define abbreviations the first time they are used in the manuscript

•Authors could check for grammatical errors through the document.

6. PLOS authors have the option to publish the peer review history of their article (what does this mean?). If published, this will include your full peer review and any attached files.

Reviewer #1: No

Reviewer #2: **Yes: **Steven Wambua

---

## [Author Response · Author response to Decision Letter 0]

20 Oct 2023

Dear Sir/Madam,

I hope this email finds you well. I wanted to take a moment to express my sincere gratitude for the time and effort you invested in reviewing and editing our study. Your invaluable input has significantly contributed to the enhancement of our study's quality, and we truly appreciate your dedication to advancing the field.

Your comprehensive feedback and critical insights provided in your review were instrumental in refining our research and addressing its weaknesses effectively. Your expertise and thorough examination of the study have undoubtedly elevated its academic rigor and credibility.

I would like to inform you that we have thoroughly reviewed your comments and suggestions, and we have made the necessary revisions accordingly. To facilitate this process, we have uploaded the revised manuscript as per your recommendations as well as a response file. You can access the updated version by following the link provided in the upload files.

Your commitment to scholarly excellence and your willingness to collaborate in the peer-review process are deeply appreciated. It is individuals like you who make valuable contributions to the academic community and help us strive for excellence in our field.

Once again, thank you for your time, expertise, and dedication to improving our study. We look forward to your feedback on the revised manuscript and hope that it aligns with your expectations.

If you have any further comments or require additional information, please do not hesitate to reach out to us. Your insights are highly valued and will undoubtedly help us achieve our goal of contributing to healthcare knowledge.

Thank you for being an essential part of our academic journey. We are sincerely grateful for your support and guidance.

Warm regards,

All authors

---

## [Decision Letter · Decision Letter 1]

12 Nov 2023

PONE-D-23-22693R1Impact of COVID-19 on patterns of drug utilization: a case study at national hospitalPLOS ONE

Dear Dr. Nguyen,

Thank you for submitting your manuscript to PLOS ONE. After careful consideration, we feel that it has merit but does not fully meet PLOS ONE’s publication criteria as it currently stands. Therefore, we invite you to submit a revised version of the manuscript that addresses the points raised during the review process.

We look forward to receiving your revised manuscript.

Kind regards,

Chinh Quoc Luong, MD., PhD.

Academic Editor

PLOS ONE

Journal Requirements:

Reviewers' comments:

Reviewer's Responses to Questions

**Comments to the Author**

1. If the authors have adequately addressed your comments raised in a previous round of review and you feel that this manuscript is now acceptable for publication, you may indicate that here to bypass the “Comments to the Author” section, enter your conflict of interest statement in the “Confidential to Editor” section, and submit your "Accept" recommendation.

Reviewer #1: (No Response)

Reviewer #2: All comments have been addressed

2. Is the manuscript technically sound, and do the data support the conclusions?

Reviewer #1: Yes

Reviewer #2: Yes

3. Has the statistical analysis been performed appropriately and rigorously? 

Reviewer #1: Yes

Reviewer #2: Yes

4. Have the authors made all data underlying the findings in their manuscript fully available?

Reviewer #1: No

Reviewer #2: Yes

5. Is the manuscript presented in an intelligible fashion and written in standard English?

Reviewer #1: Yes

Reviewer #2: Yes

6. Review Comments to the Author

Reviewer #1: Thank you for the opportunity to read this revision. For continuity purposes, I was Reviewer 1 in the previous round.

As with the previous version, the authors employ data on the prescriptions dispensed from a national geriatric hospital in Ho Chi Minh City, Viet Nam to evaluate how the Covid-19 pandemic and associated policy responses impacted drug utilization. My opinion regarding the obvious importance and relevance of this paper is unchanged from the last round—the paper addresses a valuable question.

The authors find that there was a significant surge in the number of prescriptions dispensed and the quantity of drugs dispensed. There was some heterogeneity in the types of drugs that saw such a surge; overall, the patterns were consistent with “stocking up” and didn’t drive long term increases in utilization.

In this revision, the authors have largely addressed most of my concerns. I do think some additional discussion is warranted around the dependent variable (DDD/1000P) especially in light of the clarification around median age (Reviewer 2 Comment 1).

Major Comments:

Selection on Age and Comorbidities

It is noted that “The median age of the patients were 58 in Period 1 and 67 years old in Period 2.” This seems like a big change—it is understandable that Covid could shift the demographics of the patients and that this could then explain the results (especially the heterogeneity in drug use patterns, like increased use of cardiovascular drugs). We see similar shifts in the number of diseases diagnosed on average changing from Period 1 to Period 2, and the decreased number of prescriptions between the two periods across categories.

This ties into the dependent variable…I largely accept the explanation provided in response to my previous comment (Reviewer 1 Comment 4).

I think the paper would benefit from some contextualization (maybe in the discussion section) to put some perspective around the changes…it looks like the changes in utilization were really driven by the patient population being served. This has important implications for the conclusion statements and key takeaways (e.g., “COVID-19 resulted in a sudden, non-significant increase in overall drug consumption levels among outpatients.”). It doesn’t look (to me) as if suddenly people needed more cardiovascular drugs…this result is driven by the patients who are still being served.

I don’t think this compromises the results but it does add some important perspective—for the clinicians still serving the non-Covid-19 patients, their patients will-on average-be much sicker.

Reviewer #2: One of my comments was not addressed by the author. That is "Could the authors add on Table 1 the missing data for each characteristic to help the reader

know what proportion of data was missing for each characteristic as I understand the analysis

was done for complete data a.k.a Complete Case Analysis (CCA)."

7. PLOS authors have the option to publish the peer review history of their article (what does this mean?). If published, this will include your full peer review and any attached files.

Reviewer #1: No

Reviewer #2: No

---

## [Author Response · Author response to Decision Letter 1]

6 Dec 2023

Reviewer 1:

I think the paper would benefit from some contextualization (maybe in the discussion section) to put some perspective around the changes…it looks like the changes in utilization were really driven by the patient population being served. This has important implications for the conclusion statements and key takeaways (e.g., “COVID-19 resulted in a sudden, non-significant increase in overall drug consumption levels among outpatients.”). It doesn’t look (to me) as if suddenly people needed more cardiovascular drugs…this result is driven by the patients who are still being served. I don’t think this compromises the results but it does add some important perspective—for the clinicians still serving the non-Covid-19 patients, their patients will-on average-be much sicker.

Response: We really appreciate that you pointed out the issue. After revising the paper, we think that it is crucial to complement information about the outpatients’ context in the hospital. 

Regarding this hospital, most of patients claim their prescription with the social health insurance, and it can only be done in the registered hospital in the insurance. There was a large proportion of chronic diseases observed in the study population, which indicated that these outpatients have been examined in this hospital for a while. Besides, increases in the drug consumption in the period 2 were mostly driven by the surge of the consumption observed in the initial stage of the pandemic outbreak, which illustrated the stockpiling behavior, and thus supporting the abovementioned indication. Therefore, we can anticipate that there were few first-time outpatients with chronic diseases as the majority of patients has the long-term follow-up at the hospital. We believe that this is equivalent to “the patient population being served” in your opinion. We have added the comments with respect to the outpatients in the discussion. 

Reviewer 2:

Could the authors add on Table 1 the missing data for each characteristic to help the reader

know what proportion of data was missing for each characteristic as I understand the analysis

was done for complete data a.k.a Complete Case Analysis (CCA).

Response: We sincerely apologize that this comment was missed out during the first revision. In terms of the first step of excluding missing data, missing data mainly encompassed the omitted values of age and diagnosed diseases. Therefore, we were unable to aggregate such missing values into each category of characteristics in the table. We think that missing data excluded did not affect the analysis of the finalized data.

---

## [Decision Letter · Decision Letter 2]

2 Jan 2024

Impact of COVID-19 on patterns of drug utilization: a case study at national hospital

PONE-D-23-22693R2

Dear Dr. Nguyen,

We’re pleased to inform you that your manuscript has been judged scientifically suitable for publication and will be formally accepted for publication once it meets all outstanding technical requirements.

Kind regards,

Assoc. Prof. Chinh Quoc Luong, MD., PhD.

Academic Editor

PLOS ONE

Additional Editor Comments (optional):

Reviewers' comments:

Reviewer's Responses to Questions

**Comments to the Author**

1. If the authors have adequately addressed your comments raised in a previous round of review and you feel that this manuscript is now acceptable for publication, you may indicate that here to bypass the “Comments to the Author” section, enter your conflict of interest statement in the “Confidential to Editor” section, and submit your "Accept" recommendation.

Reviewer #1: All comments have been addressed

Reviewer #2: All comments have been addressed

2. Is the manuscript technically sound, and do the data support the conclusions?

Reviewer #1: Yes

Reviewer #2: Yes

3. Has the statistical analysis been performed appropriately and rigorously? 

Reviewer #1: Yes

Reviewer #2: Yes

4. Have the authors made all data underlying the findings in their manuscript fully available?

Reviewer #1: No

Reviewer #2: Yes

5. Is the manuscript presented in an intelligible fashion and written in standard English?

Reviewer #1: Yes

Reviewer #2: Yes

6. Review Comments to the Author

Reviewer #1: Thank you for your effort in this revision. This paper forms a valuable part of the record of what happened in the early days of COVID-19 and is consistent with some of the evidence in other countries. It could inform the public health response for future pandemics.

Reviewer #2: (No Response)

7. PLOS authors have the option to publish the peer review history of their article (what does this mean?). If published, this will include your full peer review and any attached files.

Reviewer #1: **Yes: **Jeffrey David Clement

Reviewer #2: **Yes: **Steven Wambua

---

## [Editor Report · Acceptance letter]

11 Jan 2024

PONE-D-23-22693R2 

PLOS ONE

Dear Dr. Nguyen-Thi, 

I'm pleased to inform you that your manuscript has been deemed suitable for publication in PLOS ONE. Congratulations! Your manuscript is now being handed over to our production team.

Kind regards, 

on behalf of

Assoc. Prof. Chinh Quoc Luong 

Academic Editor

PLOS ONE